# Vancomycin Area under the Concentration-Time Curve Estimation Using Bayesian Modeling versus First-Order Pharmacokinetic Equations: A Quasi-Experimental Study

**DOI:** 10.3390/antibiotics11091239

**Published:** 2022-09-13

**Authors:** Yazed Saleh Alsowaida, David W. Kubiak, Brandon Dionne, Mary P. Kovacevic, Jeffrey C. Pearson

**Affiliations:** 1Department of Clinical Pharmacy, College of Pharmacy, Hail University, Hail 81442, Saudi Arabia; 2Department of Pharmacy Services, Brigham and Women’s Hospital, Boston, MA 02115, USA; dwkubiak@bwh.harvard.edu (D.W.K.); bdionne@bwh.harvard.edu (B.D.); mkovacevic@bwh.harvard.edu (M.P.K.); jcpearson@bwh.harvard.edu (J.C.P.); 3Department of Pharmacy and Health Systems Sciences, School of Pharmacy and Pharmaceutical Sciences, Bouvé College of Health Sciences, Northeastern University, Boston, MA 02115, USA

**Keywords:** vancomycin, AUC, Bayesian, therapeutic drug monitoring

## Abstract

Aim: To evaluate the efficiency of Bayesian modeling software and first-order pharmacokinetic (PK) equations to calculate vancomycin area under the concentration-time curve (AUC) estimations. Methods: Unblinded, crossover, quasi-experimental study at a tertiary care hospital for patients receiving intravenous vancomycin. Vancomycin AUC monitoring was compared using Bayesian modeling software or first-order PK equations. The primary endpoint was the time taken to estimate the AUC and determine regimen adjustments. Secondary endpoints included the percentage of vancomycin concentrations usable for AUC calculations and acute kidney injury (AKI). Results: Of the 124 patients screened, 34 patients had usable vancomycin concentrations that led to 44 AUC estimations. Without electronic health record (EHR) integration, the time from assessment to intervention in the Bayesian modeling platform was a median of 9.3 min (quartiles Q_1_–Q_3_ 7.8–12.4) compared to 6.8 min (Q_1_–Q_3_ 4.8–8.0) in the PK equations group (*p* = 0.004). With simulated Bayesian software integration into the EHR, however, the median time was 3.8 min (Q_1_–Q_3_ 2.3–6.9, *p* = 0.019). Vancomycin concentrations were usable in 88.2% in the Bayesian group compared to 48.3% in the PK equation group and there were no cases of AKI. Conclusion: Without EHR integration, Bayesian software was more time-consuming to assess vancomycin dosing than PK equations. With simulated integration, however, Bayesian software was more time efficient. In addition, vancomycin concentrations were more likely to be usable for calculations in the Bayesian group.

## 1. Introduction

Vancomycin is recommended first-line for the treatment of methicillin-resistant *Staphylococcus aureus* (MRSA) infections [1]. The vancomycin pharmacodynamic parameter most closely associated with efficacy is the area under the concentration-time curve to minimum inhibitory concentration (AUC:MIC) ratio [2,3]. Consensus 2009 vancomycin monitoring guidelines recommended targeting serum trough concentration of 15–20 μg/mL for complicated MRSA infections as a surrogate to achieve an AUC:MIC ratio of >400 [4]. Historically, trough serum vancomycin concentrations were considered the most practical method to monitor the effectiveness and safety of systemic vancomycin. However, accumulating literature has revealed that trough serum vancomycin concentrations are not optimal surrogates for estimating the target AUC:MIC ratio [2,5]. Dosing vancomycin using trough concentrations has been associated with an increase in the incidence of nephrotoxicity [2]. In contrast, monitoring vancomycin via AUC estimations is associated with a lower frequency of nephrotoxicity [6,7].

Due to evolving vancomycin pharmacokinetic/pharmacodynamic (PK/PD) and safety data and the availability of more practical methods to estimate vancomycin AUCs, updated vancomycin monitoring guidelines now endorse AUC-guided dosing and monitoring as the preferred method when treating serious MRSA infections [2]. To maximize vancomycin efficacy and minimize the risk of toxicity, the recommended target AUC:MIC ratio is 400–600.

Vancomycin AUC estimations can be calculated using either Bayesian modeling software or first-order pharmacokinetic (PK) equations [2]. The Bayesian approach relies on rigorous population vancomycin PK models to predict vancomycin AUC using patient-specific PK data. Notably, the Bayesian method may only require one vancomycin concentration drawn any time after the vancomycin infusion completes, which may result in fewer needed resources for vancomycin AUC monitoring. On the other hand, the first-order PK equations require two appropriately timed serum vancomycin concentrations. While both approaches have strengths and limitations, Bayesian methods are preferred by current vancomycin monitoring guidelines. Previous literature has demonstrated that Bayesian estimations correlate reasonably well with first-order kinetic estimates [8,9]. However, it remains unclear which of these two methods is more feasible to integrate into pharmacists’ daily workflow. Important factors for pharmacy efficiency include time commitment for therapeutic drug monitoring (TDM), the hospital’s financial resources, and the overall workload [10]. The objective of this study was to evaluate the efficiency of these two approaches for estimating a vancomycin AUC to guide vancomycin dosing in clinical practice.

## 2. Materials and Methods

### 2.1. Design, Setting, and Participants

This was an unblinded, single-center, crossover, quasi-experimental study conducted in four inpatient medical/surgical units at Brigham and Women’s Hospital. The study was granted exemption from the Institutional Review Board (protocol number 2019P003003) as it was deemed a quality improvement project, which also includes the dissemination of the study’s results as a published journal article. The study was conducted between November 2019 and January 2020 for a total of 64 days; all patients admitted to the four specified units who received systemic vancomycin were screened for inclusion in the study. For the first month, two medical units were assigned to the Bayesian group, while two surgical units were assigned to the PK equations group. Then, the vancomycin AUC assessment method was switched in the second month of the study. Patients were excluded if they were receiving vancomycin for surgical prophylaxis, were on hemodialysis, if vancomycin was being dosed based on random concentrations due to impaired renal function, or if vancomycin concentrations were never collected.

### 2.2. Education of Healthcare Personnel

Prior to the intervention period, the research team (four pharmacists and a pharmacy resident) educated all necessary healthcare providers working in the designated units, including physicians, nursing staff, phlebotomy, and pharmacists, to reduce the likelihood that vancomycin concentrations ordered by the study team would be re-timed or canceled. Education consisted of live presentations at staff meetings, educational emails, and posters displayed in the study patient care units. Examples of the educational material used can be found in the Appendix A.

### 2.3. Study Procedure

The research team was notified of new vancomycin orders or concentration results using surveillance software (TheraDoc^®^ version 4.7.5) daily between 7 AM and 9 PM. InsightRX^®^ (InsightRX^®^ NOVA version, San Francisco, CA, USA) was used to perform the Bayesian modeling calculations. The Thomson model was used for non-obese patients, while the Carreno model was used for obese patients [11,12]. The Bayesian platform utilized in the study was web-based; however, some Bayesian programs can also be integrated into the EHR to save on data input time and prevent transcription errors [13,14,15]. At the time of our study, the Bayesian modeling software was not integrated into our EHR. The first-order PK equations were performed using the Kinetics Navigator embedded in the Epic^®^ EHR, which requires manual entry of data to generate an AUC estimation. The time required by the research team to evaluate vancomycin concentrations and calculate the AUC was recorded using Toggl (Toggl, Tallinn, Estonia). Following the evaluation for inclusion, the research team coordinated with the primary team clinicians to order and time all vancomycin doses and concentrations as appropriate. The full research protocol can be found in the Appendix A.

### 2.4. Vancomycin Samples

The AUC calculation in the Bayesian group used a single random vancomycin concentration drawn with morning labs. In contrast, in the first-order PK equations group, two concentrations were drawn at the steady state (after at least 3 vancomycin doses) four hours after the start of infusion (random) and one hour prior to the next dose (trough). At our institution, vancomycin infused peripherally was administered over two to three hours to mitigate phlebitis, so four hours was chosen as the time for random concentrations to be drawn to prevent potential lab draws while vancomycin was still infusing.

### 2.5. Study Endpoints

The major endpoint of this analysis was the time taken by the research team to calculate a patient-specific vancomycin AUC and determine necessary regimen modifications if the AUC was outside of the target range of 400–600 mg∙h/L with the Bayesian modeling platform (Figure 1) compared to first-order PK equations (Figure 2). A single patient may have multiple vancomycin AUC assessments counting towards this primary endpoint. In the Bayesian group, the time measurement was separated into two phases: the time from concentration review to completion of data transcription in the Bayesian software and the time from data transcription completion to intervention documentation in the EHR. This distinction was made to account for simulated Bayesian platform integration into EHR since the research team was using a web-based Bayesian platform. If the Bayesian platform was embedded into the EHR, there would be no transcription time, as that data would automatically populate into the application. For the first-order PK equations, time was calculated from concentration review to intervention documentation. Secondary endpoints included the time from the intervention to new vancomycin order change, number of vancomycin concentrations drawn per vancomycin-day, percentage of vancomycin concentrations that could be used to calculate the AUC (i.e., for 2-level monitoring, two concentrations drawn within the same dosing interval and not during the vancomycin infusion), number of vancomycin dose changes per vancomycin-day, and incidence of acute kidney injury (AKI). AKI was defined using the Acute Kidney Injury Network classification criteria (absolute increase in SCr of ≥0.3 mg/dL, percentage increase in SCr ≥ 50%, or a decrease in urine output < 0.5 mL/kg/h for ≥6 h within a 48-h period) [16]. Our laboratory uses the Jaffe method to calculate serum creatinine.

### 2.6. Post-Hoc Analysis

A post hoc comparison of AUC estimates for both Bayesian and first-order PK equations was conducted for all patients with two concentrations drawn in the same vancomycin interval. The accuracy (defined as a ratio of estimated AUC Bayesian to AUC PK) was determined using both levels, just the first level, or just the second level, compared to the reference point of two-level PK calculations as the denominator, similar to what had been done previously by Turner and colleagues [8].

### 2.7. Statistical Analysis

Descriptive statistics were used to summarize patient demographics and clinical outcomes, presented as medians and quartiles (Q_1_–Q_3_) or percentages, as appropriate. The Chi-square test was used for categorical data, and the Mann–Whitney U test was used for continuous, non-parametric data. All statistical tests were performed in IBM^®^ SPSS Statistics (Armonk, New York, NY, USA), version 25. A *p*-value < 0.05 was considered statistically significant.

## 3. Results

### 3.1. Patient Characteristics

A total of 124 patients received vancomycin during the study period, 47 of whom were included in the analysis: 28 patients in the Bayesian group and 19 patients in the first-order PK equations group. Notably, fewer patients with assessable concentrations were included in the first-order PK equations group due to a lower number of usable vancomycin concentrations (Figure 3). Patients were primarily excluded if they were receiving vancomycin for surgical prophylaxis (*n* = 40) or never had vancomycin concentrations drawn (*n* = 32). Patient demographics and baseline characteristics can be found in Table 1. A total of 34 patients had usable vancomycin concentrations leading to 44 AUC estimations: 30 in the Bayesian group and 14 in the first-order PK equations group. In the Bayesian arm, 22 assessments (73.3%) were performed using the Thomson model, while 8 assessments (26.7%) were performed using the Carreno model for obesity [11,12].

### 3.2. Vancomycin AUC Assessment

The median time to assess an AUC in the Bayesian group, including data transcription, was 9.3 min (Q_1_–Q_3_ 7.8–12.4) compared to 6.8 min (Q_1_–Q_3_ 4.8–8.0) in the first-order PK equations group (*p* = 0.004). However, in the simulated Bayesian modeling platform integrated into the EHR, the median time from assessment to intervention decreased to 3.8 min (Q_1_–Q_3_ 2.3–6.9, *p* = 0.019) (Table 2). Vancomycin concentrations were usable for AUC calculations in 30 of 34 instances in the Bayesian group (88.2%) compared to 28 of 58 in the first-order PK group (48.3%) (*p* = 0.001). Despite having similar numbers of vancomycin concentrations drawn per vancomycin-day (0.4), dose adjustments were made more than twice as frequently in the Bayesian group (0.19 vs. 0.07 adjustments per vancomycin-day). No patients in either cohort developed acute kidney injury.

### 3.3. Vancomycin AUC Assessment

In our post hoc analysis directly comparing the two AUC strategies in the patients with two vancomycin concentrations drawn in the same dosing interval (*n* = 14 assessments), the median ratio of estimated AUC for two-concentration PK vs. two-concentration Bayesian was 1.05 (Q_1_–Q_3_ 0.98–1.14). When comparing two-concentration PK to single-concentration Bayesian, the median ratios were 1.01 (Q_1_–Q_3_ 0.97–1.04) when using the first concentration only and 0.99 (Q_1_–Q_3_ 0.95–1.01) when using the second concentration only. In 14% of assessments (2/14), the two-concentration Bayesian was outside the AUC range of 400–600 mg∙h/L when the two-concentration PK was at goal; this discrepancy occurred once if using the first concentration only and did not occur when using the second concentration only (Appendix A).

## 4. Discussion

Our study provides valuable considerations when deciding how to best implement AUC-based monitoring for vancomycin. In this study, the simulated Bayesian modeling platform with EHR integration took less time to estimate a vancomycin AUC and recommend dose adjustments compared to first-order PK equations with the Epic^®^ PK Navigator. The primary reason for the difference was that the Epic^®^ PK Navigator requires users to input data on age, weight, dosing, and concentrations for each assessment. This necessary data entry can also be subject to transcription errors that would be avoided with an integrated Bayesian modeling platform, which automatically populates patient-specific data from the EHR. If the Bayesian software is not integrated into the EHR, however, it takes more time than PK equations because additional information input is required (e.g., every vancomycin dose, serum creatinine measurement, and previous vancomycin concentration).

Our findings on time to estimate a vancomycin AUC using different methods add to a recent study by Chung and colleagues, which evaluated pharmacist time dedicated to the vancomycin dosing [10]. The authors found that the median amount of time dedicated to vancomycin-related activities per 8-h weekday shift was 10.45 min (Q_1_–Q_3_ 6.94–15.8), in which trough-based dosing was utilized for the vast majority of patients (~87%). While the overall median time requirement per vancomycin assessment was 3.45 min, vancomycin consults that utilized AUC-based monitoring required more time per patient than trough-based monitoring (median 6.2 min compared to 3.48 min). Vancomycin AUC time calculation may have an indirect effect on patients’ health since vancomycin TDM is a significant responsibility for hospital pharmacists. Thus, optimizing the time commitment for vancomycin AUC can allow pharmacists to have additional time for patient care activities, which may indirectly improve patients’ outcomes.

In our study, vancomycin concentrations from the first-order PK equations group often resulted in unusable concentrations for AUC calculations, primarily because only one was collected when two were necessary (Figure 3). This occurred even though the research team was actively ordering, monitoring, and assessing all vancomycin concentrations during the study period, which may not be feasible for front-line clinical pharmacists or clinicians. The greater flexibility with the timing of vancomycin concentrations in the Bayesian group led to only three unusable concentrations, all of which were due to the discontinuation of vancomycin therapy prior to concentration assessment. This flexibility resulted in a greater number of interventions, vancomycin AUC assessments, and more than twice the number of vancomycin dose changes per patient per vancomycin treatment day.

Bayesian dosing may allow regimens not achieving the target AUC range of 400–600 mg∙h/L to be adjusted earlier in the treatment course. Conversely, Bayesian dosing may also result in unnecessary early concentrations being drawn when vancomycin may be discontinued early in the course (within 24–48 h). Balancing early monitoring with the likelihood of needing to continue vancomycin as definitive therapy (i.e., for MRSA) is an important consideration to minimize excess vancomycin concentrations being drawn. In our study, over half of the patients were excluded due to receiving vancomycin for surgical prophylaxis or only receiving a short course of empiric vancomycin in which no vancomycin concentrations were drawn. It is important to note that these patients do not require vancomycin AUC monitoring with Bayesian dosing or PK equations.

First-order PK equations assume that the patient is at a steady state. In our post-hoc analysis, the Bayesian predicted AUC was estimated to be significantly higher in two patients compared to the PK-derived AUC, but the Bayesian estimated AUC at the time of the concentrations being drawn matched the PK-derived AUC, suggesting that these patients were not yet at steady state despite having received three vancomycin doses. This could potentially lead to underestimation of AUC and inappropriate dose increases when using first-order equations after just three doses of vancomycin.

A potential concern with Bayesian monitoring is that a single concentration may not be sufficient to provide an accurate AUC estimation. In our study, the single-concentration AUC estimates were generally within five percent of two-concentration Bayesian estimates and rarely conflicted with two-concentration PK in terms of if the AUC was within the goal range, suggesting that a single-concentration strategy may be feasible. The accuracy of calculating a vancomycin AUC by a Bayesian approach measured against first-order PK equations has initially evaluated by Turner and colleagues [8]. They compared five Bayesian dosing platforms to two first-order PK equations, sampling numerous vancomycin concentrations in 19 adult patients. Overall, the accuracy of Bayesian modeling using trough-only monitoring was comparable to 2-concentration first-order PK equations, with the median ratio of accuracy ranging from 0.79 to 1.03. In their study, InsightRX^®^ had a median accuracy (Q_1_–Q_3_) of 0.84 (0.77–0.88), which was average compared to the other models assessed. A similar finding was published using PrecisePK Bayesian modeling software for 65 AUC calculations [17].

A more recent study by Olney and colleagues evaluated and compared clinical agreement among AUC estimation with Bayesian using InsightRX^®^ with two concentrations, Bayesian with one concentration, and first-order PK equations with two concentrations [9]. While there was an excellent correlation and clinical agreement with Bayesian two concentrations and first-order PK equations (r = 0.963) with 87.4% clinical agreement, Bayesian one concentration demonstrated moderately high correlation (r = 0.823), with 76.8% clinical agreement. Yet, the correlation between Bayesian two vs. one concentration was strong (r = 0.931) with 88.5% clinical agreement. A single-concentration estimation using Bayesian modeling saves time by requiring fewer samples to be collected and analyzed, reduces the number of venipunctures, and likely also saves the hospital money [18]. Without a true “gold standard” for vancomycin AUC calculations, our study team believes that the benefits of drawing vancomycin concentrations once per assessment at regularly scheduled times outweigh the risk of suboptimal clinical agreement with first-order PK equations, especially if compared to the previous standard of trough-based monitoring.

To our knowledge, this is the first study directly comparing the feasibility of implementing vancomycin AUC monitoring by Bayesian modeling versus first-order PK equations. Implementation of vancomycin AUC monitoring into clinical practice is challenging and time-consuming, so maximizing the efficiency of the process is paramount to clinician adoption and success. Our study has several limitations, the most prominent being that it was unblinded and conducted in a single center in a relatively short period of time with a small sample size. In the first-order PK group, vancomycin assessments were performed less often due to a higher percentage of concentrations that could not be used. The primary endpoint was assessed for the Bayesian dosing group using InsightRX^®^ software, so the time endpoints cannot be extrapolated to other Bayesian modeling platforms. Bayesian modeling was also not integrated into the EHR, so adjustments in time calculations had to be accounted for to simulate EHR integration. The total study period was just two months in duration, which may have been inadequate for clinicians to become accustomed to the different vancomycin monitoring approaches. At the time of the study, vancomycin AUC monitoring was in a trial period and not established at our institution. Thus, the study population included only medical and surgical adult patients, and neither pediatric nor critically ill patients were included in the study. There was also variability in the time calculations among the study team members, with some study team members being more efficient than others in the time taken to assess concentrations and AUC in both groups. However, this reflects real-world practice as different users will have varying levels of speed and comfortability with the various platforms. Despite extensive education, there were some clinicians who deviated from the study protocol, requiring additional education. Lastly, we did not assess clinical outcomes in the study aside from AKI but instead focused on evaluating the efficiency of the two vancomycin AUC assessment methods.

## 5. Conclusions

Using Bayesian-guided vancomycin AUC monitoring as a standalone product required more time to calculate a vancomycin AUC compared to using an EHR-embedded first-order PK calculator in our study. With simulated EHR integration, however, Bayesian-guided dosing was estimated to take less time than PK equations. Additionally, the Bayesian approach resulted in more usable vancomycin concentrations for AUC calculations than the first-order PK equations. When transitioning to AUC-guided vancomycin dosing and if Bayesian models are in consideration, these findings highlight the value of EHR integration.

## Figures and Tables

**Figure 1 antibiotics-11-01239-f001:**
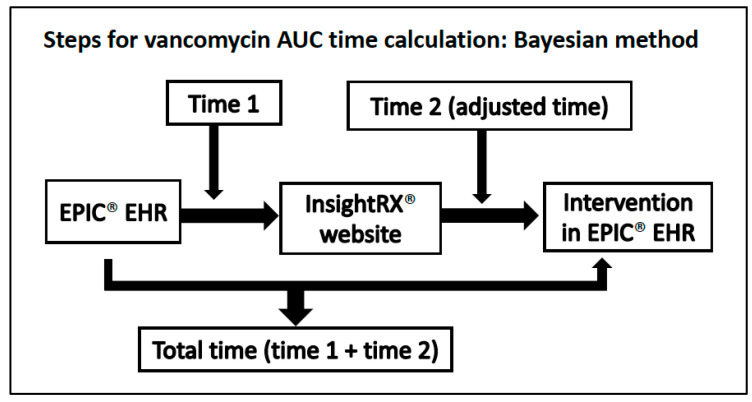
Procedure for vancomycin AUC time calculation using the Bayesian method.

**Figure 2 antibiotics-11-01239-f002:**
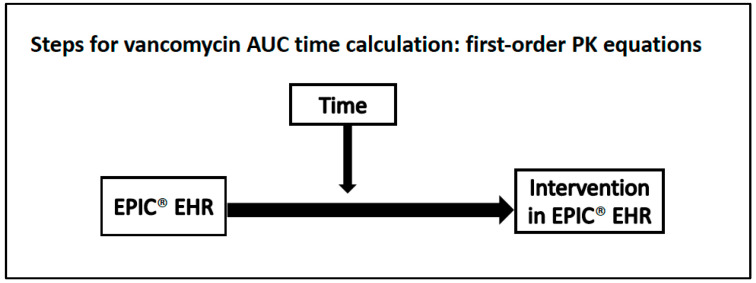
Procedure for vancomycin AUC time calculation using the first-order PK equations method.

**Figure 3 antibiotics-11-01239-f003:**
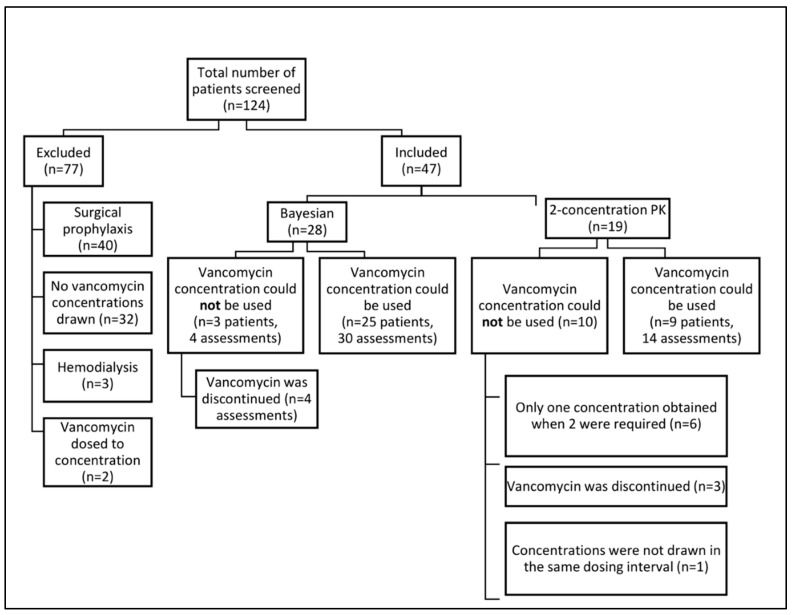
Flow diagram for patient enrollment and inclusion and exclusion criteria.

**Table 1 antibiotics-11-01239-t001:** Baseline characteristics.

Variable	Bayesian (*n* = 28)	First-Order PK Equations (*n* = 19)
Age, years	64 (47–71)	55 (41–72)
Female, *n* (%)	17 (60.7%)	14 (73.7%)
Weight, kg	79.0 (66.3–94.2)	88.9 (71.4–103.9)
Baseline serum creatinine, mg/dL	0.8 (0.7–1.0)	0.9 (0.7–1.0)
Initial vancomycin maintenance dose, mg/kg/day	30.3 (26.3–35.2)	29.4 (27.3–35.0)
Vancomycin indication, *n*	Pneumonia: 7 Skin and soft tissue infection: 6 Bloodstream infection: 5 Urinary tract infection: 3 Meningitis: 2 Sepsis: 2 Septic arthritis: 2 Intra-abdominal infection: 1 Osteomyelitits: 0	Pneumonia: 1 Skin & soft tissue infection: 7 Bloodstream infection: 3 Urinary tract infection: 1 Meningitis: 3 Sepsis: 0 Septic arthritis: 0 Intra-abdominal infection: 1 Osteomyelitis: 3
Infectious diseases consultation, *n* (%)	7 (25.0%)	8 (42.1%)

All data reported as median (Q_1_–Q_3_) unless otherwise indicated. PK = pharmacokinetic.

**Table 2 antibiotics-11-01239-t002:** Primary and secondary endpoints.

Variable	Bayesian (*n* = 30)	First-Order PK Equations (*n* = 14)	*p*-Value
**Primary endpoint**			
Total time taken to assess AUC, minutes	9.3 (7.8–12.4)	6.8 (4.8–8.0)	0.004
Adjusted time taken to assess AUC, minutes ^a^	3.8 (2.3–6.9)	6.8 (4.8–8.0)	0.019
**Secondary endpoints**			
Time from intervention to new vancomycin order, minutes	23.5 (13.5–66.5)	31 (5–74)	0.089
Vancomycin concentrations per vancomycin-day, *n* (per day of therapy)	34/95 (0.4)	58/154 (0.4)	–
Usable vancomycin concentrations to calculate AUC, *n* (%) ^b^	30/34 (88.2)	28/58 (48.3)	0.001
Dose adjustments per vancomycin-day, *n* (per day of therapy)	18/95 (0.19)	11/154 (0.07)	–
Incidence of acute kidney injury, *n*	0	0	–

All data reported as median (Q_1_–Q_3_) unless otherwise indicated, PK = pharmacokinetic, AUC = area under the concentration-time curve, ^a^: Time 2 in Figure 1 for the Bayesian group, ^b^: Reasons for unusable vancomycin concentrations can be found in Figure 2.

## Data Availability

Not applicable.

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
