# Peer review of "Vancomycin Area under the Concentration-Time Curve Estimation Using Bayesian Modeling versus First-Order Pharmacokinetic Equations: A Quasi-Experimental Study"

_antibiotics, 2022, doi:10.3390/antibiotics11091239_

Round 1
Reviewer 1 Report
Major comments:
1. Please include the vancomycin therapeutic index in the introduction so readers will know why is important to predict the AUC of vancomycin.
2. Figure 2, the actual number of subjects used for the first order PK group is n=6 which is low compared to Bayesian, how do you justify a comparison between two groups?
3. How critical is time for the patients should be explained in the discussion section, and does saving a few minutes will impact patients’ health.
4. What is the major advantage of Bayesian group vs first order PK group is not clear.
Minor comments:
1. Section 2.2, looks repetitive from abstract
Author Response
Thank you for your valuable comments, you may find our responses to your comments below:
Comment 1: Please include the vancomycin therapeutic index in the introduction so readers will know why is important to predict the AUC of vancomycin.
Response 1: information about vancomycin therapeutic PK/PD index is included in the introduction in the first paragraph as follows “The vancomycin pharmacodynamic parameter most closely associated with efficacy is the AUC: MIC ratio.” Also, we included the recommended vancomycin AUC therapeutic target in the second paragraph as follows “To maximize vancomycin efficacy and minimize the risk of toxicity, the recommended target AUC: MIC ratio is 400-600.”
Comment 2: In figure 2, the actual number of subjects used for the first order PK group is n=6 which is low compared to Bayesian, how do you justify a comparison between two groups?
Response 2: an important aspect of the study is to assess the efficiency of performing vancomycin AUC with these methods. The low number of patients with assessable levels in the first-order PK group (n=9) compared to the Bayesian group (n=25) is primarily due to the number of usable vancomycin concentrations available to calculate vancomycin AUC. The first-order PK equations method requires 2 vancomycin concentration timed appropriately as described in the methods, which was challenging for the nursing/laboratory staff and resulted in vancomycin concentrations that could not be used. These points have been highlighted in the third paragraph of the discussion.
Comment 3: How critical is time for the patients should be explained in the discussion section, and does saving a few minutes will impact patients’ health.
Response 3: thank you for your comment. Although the time for estimating vancomycin AUC does not impact patient health directly, the vancomycin AUC time calculation may have an indirect effect on the available pharmacists’ time for additional patient care activities which may improve patients’ health outcomes. We added the following sentence to the second paragraph of the discussion section. “Vancomycin AUC time calculation may have an indirect effect on patients’ health since vancomycin therapeutic drug monitoring is a significant responsibility for hospital pharmacists. Thus, optimizing the time commitment for vancomycin AUC can allow pharmacists to have additional time for patient care activities which may improve patients’ outcomes.”
Comment 4: What is the major advantage of Bayesian group vs first order PK group is not clear.
Response 4: Thank you for your feedback, the most important advantage of the Bayesian method is that it requires only 1 vancomycin level to estimate vancomycin AUC; thus, makes vancomycin AUC estimations easier. We added additional information on the Bayesian method in the introduction in the third paragraph as follows: “Notably, the Bayesian method may only require one vancomycin concentration drawn any time after the vancomycin infusion completes which may result in fewer needed resources for vancomycin AUC monitoring.” In addition, more in-depth information about the advantages of the Bayesian method is available in the discussion section in Paragraphs 3 and 4.
Minor comments:
Comment 5: Section 2.2, looks repetitive from abstract
Response 5: thank you for your feedback, these are essential data and complement each other, so it is important to include for the reader to understand how the time for vancomycin AUC estimation is calculated.
Reviewer 2 Report
In the paper “Vancomycin area under the concentration-time curve estimation using Bayesian modeling versus first-order pharmacokinetic equations: A quasi-experimental study”, authors evaluated the efficiency of Bayesian modeling software and first-order pharmacokinetic (PK) equations to calculate vancomycin area under the concentration-time curve (AUC) estimations. They conducted an unblinded, single center study screening 124 patients but only 34 patients led to 44 AUC estimations resulting in conclusions of this study. The study concluded that without EHR integration, Bayesian software was more time-consuming and that with simulated integration, it was more time efficient to assess vancomycin dosing than PK equations. Overall, the conclusions align well with the aim and results, however it is important that EHR integration plays a critical role in conclusions of this study. Authors need to emphasize this as a factor as well. The article is well written and clearly described the limitations of this study.
Few places of improvement to be considered by the authors:
1) It could be important to further establish what defines the efficiency of these models that is being tested in this study. The authors have included primary and secondary end points which although defines the efficiency partly, it would be beneficial to add background on what are the factors generally considered by clinicians/pharmacists in their daily workflow (Line 62) in choosing between the two methods and how those factors fit into the efficiency that is being tested in this study.
2) As indicated by the authors, the fact that this is a single-center study and the small number of usable concentrations compromises the ability to generalize the results. Figure 1 flow diagram clearly represented the patient enrollment. But, did the study include any children, neonates or seriously ill-patients? If not, why?
This information could be useful to further support usage of Bayesian model in children where number of sampling to test can be reduced ( please check out PMID 34145165, 31402708). May be important to discuss about critically ill patients and how and why the efficiency of model differs in them (please see PMID 33938713).
Minor comments:
1) Line 13: Background is written as aim. Need to rewrite or change the title to “aim”
2) Line 36, define abbreviation at first use
3) Line 117, any reasons to justify the discrepancy can be included in the discussion in detail
4) Lines 152 to 155 unclear. Need to re-write clearly
5) Line 215, reason for not assessing clinical outcomes in the study? Does it mean the clinical outcome was not affected by choosing between the two methods?
Author Response
Thank you for your valuable comments, you may find our responses to your comments below:
Few places of improvement to be considered by the authors:
Comment 1: It could be important to further establish what defines the efficiency of these models that is being tested in this study. The authors have included primary and secondary end points which although defining the efficiency partly, it would be beneficial to add background on what are the factors generally considered by clinicians/pharmacists in their daily workflow (Line 62) in choosing between the two methods and how those factors fit into the efficiency that is being tested in this study.
Response 1: we added a sentence in the third paragraph of the introduction about the factors considered by the pharmacist for efficiency to calculate vancomycin AUC. The sentence appears as follows “Important factors for pharmacy efficiency include time commitment for therapeutic drug monitoring (TDM), the hospital's financial resources, and the overall workload”
Comment 2: As indicated by the authors, the fact that this is a single-center study and the small number of usable concentrations compromises the ability to generalize the results. Figure 1 flow diagram clearly represented the patient enrollment. But, did the study include any children, neonates or seriously ill-patients? If not, why?
This information could be useful to further support usage of Bayesian model in children where number of sampling to test can be reduced ( please check out PMID 34145165, 31402708). May be important to discuss about critically ill patients and how and why the efficiency of model differs in them (please see PMID 33938713).
Response 2: At the time the study was conducted, the vancomycin AUC was only published as a draft, and we conducted this study as a trial. Thus, vancomycin AUC monitoring was not established at our institution and we neither included pediatric nor critically ill patients. The study was conducted on medical and surgical patient floors only. We added that to the limitation section as follows: “At the time of the study, vancomycin AUC monitoring was in a trial period and not established at our institution. Thus, the study population included only medical and surgical adult patients, and neither pediatric nor critically ill patients were included in the study”
Minor comments:
Comment 1: Line 13: Background is written as aim. Need to rewrite or change the title to “aim”
Response 1: thank you for your feedback, we changed the title to aim as requested.
Comment 2: Line 36, define abbreviation at first use
Response 2: thank you, this has been addressed.
Comment 3: Line 117, any reasons to justify the discrepancy can be included in the discussion in detail
Response 3: the issues with Bayesian one vs. 2 concentration discrepancy are discussed in the 6th paragraph of the discussion section.
Comment 4: Lines 152 to 155 unclear. Need to re-write clearly
Response 4: Thank you, this has been re-written to clarify
Comment 5: Line 215, reason for not assessing clinical outcomes in the study? Does it mean the clinical outcome was not affected by choosing between the two methods?
Response 5: the objective of our study was to evaluate the logistical and operational aspects of monitoring vancomycin AUC and not clinical outcomes. Thus, we did not include clinical outcomes for the patients included as our small sample size in the pilot would be unlikely to show differences in clinical outcomes between the two groups.
Reviewer 3 Report
1. Please revise the sections, the methods section is presented after the results section. Please update appropriately.
2. Figure 2 is confusing, it references sections 1A and 1B.
3. Must state "minimum inhibitory concentration (MIC)" in the first sentence.
4. Please add in the methods section the creatinine analysis, how it was measured and how it was used in the analysis.
5. Please specify the justification behind the distribution of subjects/clinics per study arm.
6. Please confirm that the "IRB exempt" also includes dissemination of the results of the study (publication). The authors stated that "The 221 study was granted exemption from the Institutional Review Board (protocol number 222 2019P003003) as it was deemed a quality improvement project."
7. The methods section states "The AUC calculation in the Bayesian group used a single random vancomycin 260 concentration drawn with morning labs", however Figure 2 states otherwise, pleas update or clarify.
Author Response
Thank you for your valuable comments, you may find our responses to your comments below:
Comment 1: Please revise the sections, the methods section is presented after the results section. Please update appropriately.
Response 1: we also prefer the order of the manuscript with methods presented before the results as you suggest, so we have adjusted in the revised manuscript. But just to note that the Antibiotics directions place Materials and Methods after the Discussion, which is how data is presented in various recent papers from the journal (doi.org/10.3390/antibiotics11091221, doi.org/10.3390/antibiotics11081138, and doi.org/10.3390/antibiotics11081132, for example). Nevertheless, due to reviewer and author preference, we have re-organized the manuscript to have methods presented before the results, and we have updated the figure numbers accordingly. Thank you for this suggestion.
Comment 2: Figure 2 is confusing, it references sections 1A and 1B.
Response 2: thank you for your feedback, we have split figure 2 into 2 separate figures as Figure 1 (Bayesian method) and Figure 2 (first-order PK equations) to avoid this confusion
Comment 3: Must state "minimum inhibitory concentration (MIC)" in the first sentence.
Response 3: thank you, we have corrected this
Comment 4: Please add in the methods section the creatinine analysis, how it was measured and how it was used in the analysis.
Response 4: Our laboratory uses the Jaffe method and this was used to determine AKI rates, this has been added to the methods section, thank you.
Comment 5: Please specify the justification behind the distribution of subjects/clinics per study arm.
Response 5: thank you for your feedback, the study was conducted in a crossover design. As specified in the methods section 2.1, “For the first month, two medical units were assigned to the Bayesian group while two surgical units were assigned to the PK equations group. Then, the vancomycin AUC assessment method was switched in the second month of the study.” We did not intentionally recruit more patients to one of the study arms versus other. Specifically, in the first-order PK arm, fewer vancomycin concentrations were usable, with reasons provided in Figure 3, under “Vancomycin concentrations could not be used (n= 10).” We have added a sentence to highlight this in the result section 3.1 as follows “Notably, fewer patients with assessable concentrations were included in the first-order PK equations group due to lower number of usable vancomycin concentrations.”
Comment 6: Please confirm that the "IRB exempt" also includes dissemination of the results of the study (publication). The authors stated that "The 221 study was granted exemption from the Institutional Review Board (protocol number 222 2019P003003) as it was deemed a quality improvement project."
Response 6: thank you for your feedback, we included the information that the result of the study can also be published in the method section 2.1 as follows “…which also includes the dissemination of the study’s results as a published journal article”
Comment 7: The methods section states "The AUC calculation in the Bayesian group used a single random vancomycin 260 concentration drawn with morning labs", however Figure 2 states otherwise, pleas update or clarify.
Response 7: thank you for your comment. Figure 2 states the numbers for all the patients included in the study. Please note that one patient can have multiple Bayesian AUC calculations on separate days (e.g., days 1 and 4). All Bayesian AUC calculations used only 1 vancomycin concentration for the Bayesian arm.